



# A new portable sampler of atmospheric methane for radiocarbon measurements

Giulia Zazzeri[1*], Lukas Wacker[1], Negar Haghipour[2], Philip Gautchi[1], Thomas Laemmel[3], Sönke Szidat[3], Heather Graven[4]

[1] Laboratory of Ion Beam Physics, ETH, Zurich, Switzerland. *Now at: RSE, Sviluppo Sostenibile e Fonti Energetiche, Milano, Italy
[2] Geological Institute, ETH, Zurich, Switzerland
[3] Department of Chemistry, Biochemistry and Pharmaceutical Sciences, University of Bern, Bern, Switzerland & Oeschger
Centre for Climate Change Research, University of Bern, Bern, Switzerland
[4] Department of Physics, Imperial College London, London

*Correspondence to*: Giulia Zazzeri (giulia.zazzeri@rse-web.it)

Radiocarbon ($^{14}C$) is an optimal tracer of methane emissions, as $^{14}C$ measurements enable distinguishing fossil from biogenic

methane ($CH_4$). However, $^{14}C$ measurements in atmospheric methane are still rare, mainly because of the technical challenge

of collecting enough carbon for $^{14}C$ analysis from ambient air samples. In this study we address this challenge by advancing

the system in Zazzeri et al. (2021) into a much more compact and portable sampler, and by coupling the sampler with the

MICADAS AMS system at ETH, Zurich, using a gas interface.

Here we present the new sampler setup, the assessment of the system contamination and a first inter-laboratory comparison

with the LARA AMS laboratory at the University of Bern.

With our sampling line we achieved a very low blank, 0.7 µgC compared to 5.5 µgC in Zazzeri et al. (2021), and a sample

precision of 0.9 %, comparable with other measurements techniques for $^{14}CH_4$, while reducing the sample size to 60 liters of

air. We show that this technique, with further improvements, will enable routine $^{14}CH_4$ measurements in the field for an

improved understanding of $CH_4$ sources.

## 1    1 Introduction

Understanding the methane ($CH_4$) budget and identifying methane sources have become priority to tackle global warming, as

methane is the second most important anthropogenic greenhouse gas after carbon dioxide ($CO_2$) and because the dynamics

that led to the $CH_4$ increase in the last decade have not been fully unraveled. Tracing $CH_4$ sources and monitoring mitigation

strategies are urgently needed.

$^{14}C$ measurements of atmospheric methane can advance our knowledge on methane production processes by differentiating

fossil vs biogenic sources. This is because fossil $CH_4$ is depleted in $^{14}C$, and when emitted into the atmosphere, exerts a





dilution of the $^{14}$C in atmosphere that can be quantified. However, this research field is still under-explored, as $^{14}$C measurements of atmospheric methane are challenging.

One of the main challenges is sampling enough air for $^{14}$C analysis via accelerator mass spectrometry (AMS), as the

atmospheric methane concentration is low (~2 ppm). Here we build on recent advances that have been made in the analysis of $^{14}$C in atmospheric methane. Traditionally, air was collected in pressurized cylinders using high-pressure pumps, followed by an extraction procedure in the laboratory (Eisma 1994, Townsend-Small 2012). Zazzeri et al. (2021) developed a new technique that separates methane carbon from ambient air while sampling, simplifying the transportation of collected samples in a small trap and minimizing the laboratory processing needed. The F$^{14}$C measurement precision achieved is

between 0.5 and 1.2 %, comparable to the best precisions of alternative but more lab intensive techniques. The laboratory-based system developed by Zazzeri et al. (2021) was applied in the quantification of fossil and biogenic proportions of $CH_4$ in London (Zazzeri et al. 2023). A portable system using a similar technique was demonstrated by Palonen et al. (2017), but only for samples with enriched methane concentrations of >100 ppm, e.g. for $CH_4$ emissions from wetlands. Another promising recently developed technique applies chromatographic separation of $CH_4$ from air as it requires only 60 l of

atmospheric air to be sampled in a bag (Espic et al. 2019), still achieving precisions of 1.2 %.

In this study we advance the sensitive though simple methane sampling system in Zazzeri et al. (2021) with the portability of the system of Palonen et al. (2017), requiring as little air as demonstrated by Espic et al. (2019). The result is a compact and portable system to be deployed in field campaigns. We present the technology advancement and the assessment of the system efficiency, by quantifying the amount of extraneous carbon introduced during sample preparation and ultimately the

measurement precision to be achieved. We demonstrate the method by comparing $^{14}$C measurements made by the new portable system at the Laboratory of Ion beam Physics (LIP) at ETH Zurich and by the system using bag sampling and chromatographic separation at the Laboratory for the Analysis of Radiocarbon with AMS (LARA) at the University of Bern.





## 2 Method

### 1.1 The sampling setup

a)                                                                                                b)

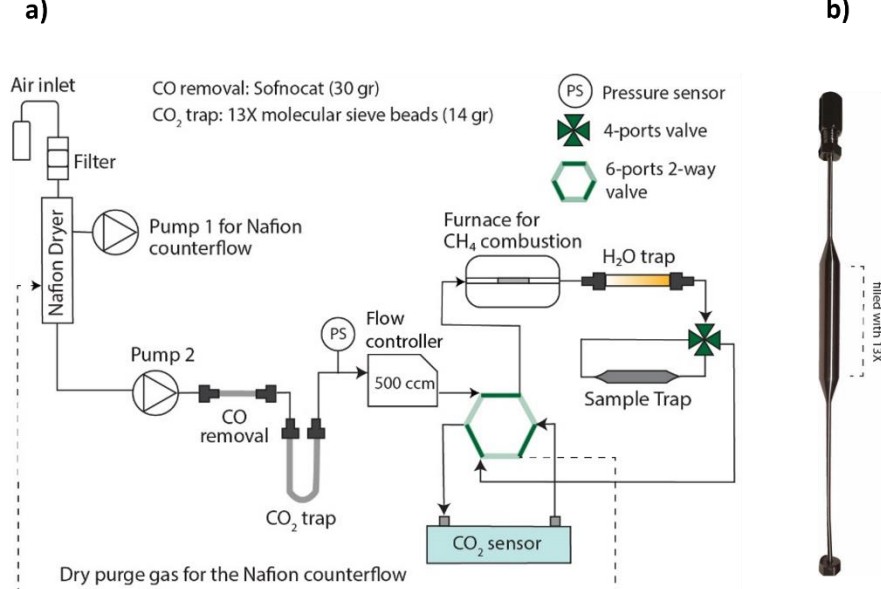

**Figure 1: a) Schematic of the sampling system. First, the filtered air is dried with a Nafion dryer; then, any CO₂ from ambient air and from oxidation of CO is removed on a trap. The CO₂ derived from the combustion of CH₄ is collected onto a final sample trap. Dark green lines in the 6-ports valve indicate active flow direction. In the indicated configuration, the CO₂ sensor measures the CO₂ level after the sample trap, enabling to check for the trap breakthrough. In the alternative configuration, complete CO₂ removal prior to CH₄ oxidation can be checked. b) Sample trap filled with 0.250 g of 13X.**

The sampling system is based on four main steps as in Zazzeri et al. (2021): 1) $H_2O$ removal with a Nafion dryer; 2) CO and $CO_2$ removal; 3) combustion of $CH_4$ to $CO_2$; 4) adsorption of the combustion-derived $CO_2$ onto a molecular sieve sample trap. Figure 1 shows the system schematic. Ambient air is sampled through a Nafion dryer at up to 500 cc/min of air with a KNF membrane pump (pump 2 in Fig 1) controlled by a mass flow controller. Downstream of the pump, CO is oxidized to $CO_2$ using Sofnocat® catalyst before all $CO_2$ (from ambient and from oxidation of CO) is removed on a trap containing 14 g of 13X molecular sieve in 1 mm pellets. This amount of molecular sieve has been found sufficient to trap atmospheric $CO_2$ in ~300 L of air. After collection of three samples, this trap is disconnected from the system via two Swagelok ball valves, then removed and regenerated by heating at 500°C with high purity nitrogen back flush for at least three hours, in a similar manner as in Zazzeri et al (2021).

After the sample air passes through the $CO_2$ trap, $CH_4$ is combusted at 800 °C in a small furnace comprising a 22 cm long quartz tube with 1 g of platinized quartz wool (Sigma Aldrich) acting as catalyst. The $CO_2$ derived from combustion of $CH_4$ is collected on the sample trap (13X, 45-60 mesh) for subsequent [14]C measurement. A non-dispersive infrared $CO_2$ sensor (NDIR FLOW[EVO] from SmartGas) monitors both completeness of $CO_2$ removal from air prior to methane combustion and completeness of $CO_2$ collection on the sample trap. If regularly calibrated and run at constant temperature and pressure, the





sensor can measure $CO_2$ concentrations in a range of 0 to 100 ppm with a precision of ±1 ppm. The sample trap, minimized

in size for low cross contamination, can be cooled with Peltier coolers to maximize trapping efficiency and avoid sample

loss.

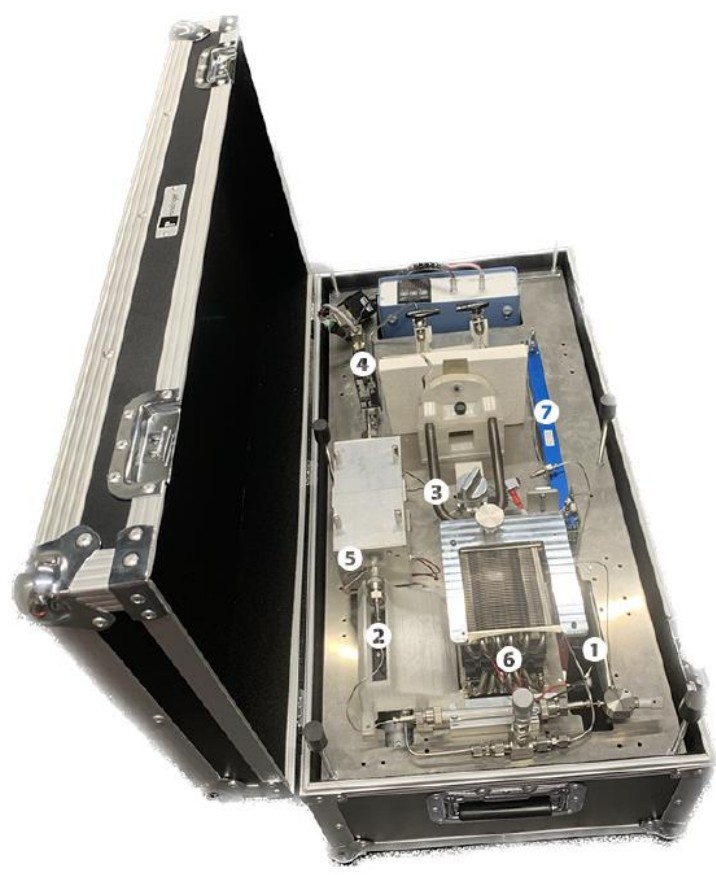

**Figure 2: Setup of the atmospheric methane sampling device. The whole system fits well into a box of 80 x 40 x 30 cm. It runs**
**either on 115/230 V AC or 48 V DC provided by a battery pack. Major parts of the system are: 1) Nafion dryer (mostly hidden**
**underneath); 2) Pumps (partially hidden); 3) $CO_2$ trap; 4) Flow controller; 5) Furnace for $CH_4$ combustion; 6) Sample trap with**
**Peltier coolers; 7) $CO_2$ sensor**

## 1.2   Sample trap and cooling system

The sample trap consists of 0.250 g 13X 45-60 mesh molecular sieve packed in a 4 cm long ¼" OD stainless steel tube. The

trap tube is welded to stainless steel capillary tubing and attached to a VICI 4 port valve which can be disconnected from the

sampling system (Fig 1 b) in order to release the sample for $^{14}C$ analysis in the AMS. Before its first use, the sample trap is

heated gradually to 650 °C in a customized oven, while flushing with high purity nitrogen. The NDIR FLOW$^{EVO}$ $CO_2$ sensor

is used to check when the trap is not releasing $CO_2$ anymore and does not contain any residual carbon, typically after 1 hour.





During sampling, the sample trap is cooled down to -10 °C using two Peltier elements, part 6 in Fig 2. This maximizes the
     trapping efficiency such that 0.250 g 13X can adsorb ~60 µgC (methane carbon from ~60 L of ambient air at 2 ppm), before
     the $CO_2$ breakthrough happens. The adsorption capacity can be enhanced by lowering the temperature even further.

     After sampling, the sample trap is disconnected from the system and heated at 450 °C for 10 minutes for sample desorption.
     The desorbed $CO_2$ is cryogenically sealed into a glass ampule to be used in the gas interface system of the Mini Carbon
Dating AMS system (MICADAS) (Wacker et al. 2013).

     Before the next sample is collected, the sample trap is cleaned of remaining $CO_2$ by flushing with high purity nitrogen while
     heating at 550 °C for 30 minutes. Such a long procedure compared to other cleaning processes for other applications is only
     precautionary, a shorter procedure might be sufficient to remove any residual carbon from previous sampling.

### 1.3    AMS analysis

The sample is measured with the MICADAS accelerator mass spectrometry facilities for radiocarbon measurements in the
     Laboratory of Ion Beam Physics ETH (Wacker et al. 2010). The $^{14}C$ analysis using the gas interface of the MICADAS takes
     about 20 minutes and achieves a measurement precision of less than 1% for modern samples (Wacker et al. 2013) for a
     sample containing 20 µg carbon. Precisions down to 0.5% can be achieved, when measurements are repeated on ≥50 µg of
     carbon (Fahrni et al. 2010).

The combusted NOX standard (SRM-4990C, Man 1983, Wacker et al. 2019) and $^{14}C$-free $CO_2$ pre-mixed with helium in gas
     bottles were measured for standard normalization respectively or blank correction. Measured data were evaluated with the
     BATS program, where the samples were fractionation corrected, blank subtracted and normalized with the NOX standard
     (Wacker et al. 2010) to obtain $F^{14}C$ values (Reimer et al. 2004).

### 1.4    Characterization of the extraneous contaminant carbon within the sampler

Extraneous contaminant carbon in the sample trap after a sample collection might derive from intrusion of lab air into the
     system, from incomplete removal of atmospheric $CO_2$, from residual carbon on the sample trap prior to sampling, or from
     impurities within the combustion column.

      To check for and quantify the contaminant carbon, we collected $CH_4$ samples of different sizes, and we follow the
     relationship between the measured fraction Modern ($F^{14}C$) versus the sample masses given by the mass balance in Eq 1:

$$F^{14}C_{meas} = F^{14}C_{true} + \frac{1}{\mu gC_{meas}}[\mu gC_{add} * (F^{14}C_{add} - F^{14}C_{true})]$$

115                                                                                                                          (1)

     where *"meas"* indicates the measured value, $F^{14}C_{true}$ the $F^{14}C$ value of the sampled air, $\mu gC_{add}$ the carbon added into the
     system and $F^{14}C_{add}$ its $F^{14}C$ value. If assuming a constant contamination, the contaminant carbon addedd to the system is
     given by the $\mu gC_{add}$ value that produces the best fitting curve through the $F^{14}C_{meas}$ values plotted against the measured
     sample masses ($\mu gC_{meas}$). We assess the goodness of fit using reduced chi-squared statistics.





To quantify the modern contaminant carbon, we collected seven samples from 10 to 70 μgC from a 2 ppm mixture of fossil methane and synthetic air with no $CO_2$, CO or hydrocarbons (Fossil Ref).

To check for any fossil contaminant, we collected seven samples, from 10 to 75 μgC, from a cylinder of pressurized ambient air (Ref 1), with a $CH_4$ mole fraction of 2040 ppb. Note that in this case, the $F^{14}C$ of the reference gas ($F^{14}C_{true}$) is unknown. The amount of fossil contaminant carbon and the $F^{14}C$ value of the reference gas are calculated by tweaking $\mu gC_{add}$ and

$F^{14}C_{true}$ in Eq 1 to produce the best fitting curve.

In order to verify the source of the contaminant carbon, we collected five blanks. Three blanks were collected by running the system with lab air and without combustion, for three hours. This enabled verification of any contaminant carbon deriving from atmospheric $CO_2$ that was not trapped in the $CO_2$ trap and from residual carbon in the sample trap. Two samples were collected by flushing the system with nitrogen and with the combustion furnace at 800 °C to verify that additional carbon

was not produced within the combustion process. No carbon was extracted from these five blanks.

## 1.5 Comparison with chromatographic extraction procedure

Three samples transferred in sampling bags from the cylinder of pressurized ambient air Ref 1 were extracted at the LARA laboratory, University of Bern, using 60 L of air and following the chromatographic extraction procedure in Espic et al. (2019). $CO_2$ derived from the sample extraction in Bern was measured using the gas interface system of the MICADAS

AMS system at ETH, in the same manner as the samples collected with our portable sampler.

## 2 Results

Table 1 shows the $F^{14}C$ values and masses of the samples collected.

| ETH nr. | Mass ugC | $F^{14}C$ | +- (%) |
|---|---|---|---|
| *Modern Samples* | | | |
| 133113.15.1 | 6 | 1.2903 | 2.27 |
| 133113.16.1 | 15 | 1.3570 | 1.03 |
| 133113.17.1 | 40 | 1.3610 | 0.76 |
| 133113.18.1 | 12 | 1.3293 | 1.10 |
| 133113.19.1 | 28 | 1.3577 | 0.85 |
| 133113.20.1 | 48 | 1.3604 | 0.85 |
| 133113.22.1 | 75 | 1.3816 | 0.82 |
| *Fossil samples* | | | |
| 136294.7.1 | 56 | 0.020612 | 6.81 |
| 136294.8.1 | 66 | 0.019753 | 5.90 |
| 136294.10.1 | 20 | 0.033567 | 5.03 |
| 136294.11.1 | 18 | 0.036063 | 5.09 |



| | | | |
|---|---|---|---|
| 136294.12.1 | 30 | 0.02058 | 5.60 |
| 136294.13.1 | 47 | 0.01954 | 5.48 |
| 136294.17.1 | 15 | 0.040006 | 4.50 |
| *Bern* | | | |
| 133991.1.1 | 66 | 1.3715 | 0.89 |
| 133991.2.1 | 64 | 1.3743 | 0.96 |
| 133991.3.1 | 66 | 1.3616 | 0.96 |

**Table 1: Mass, F¹⁴C values and uncertainty of samples collected. Modern samples are collected from Ref 1, fossil samples from Fossil Ref and "Bern" are samples extracted following the chromatographic procedure at LARA.**

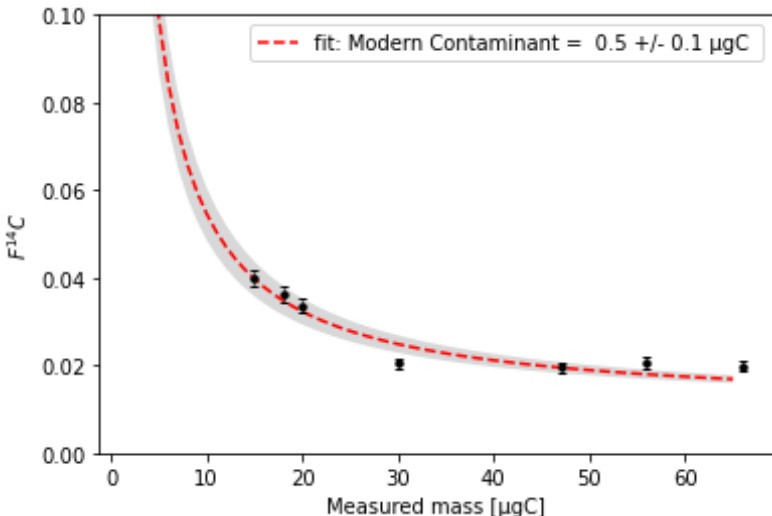

**Figure 3: F¹⁴C values against the measured mass of the samples collected from the Fossil Ref. The grey bands represent one sigma uncertainty bar on the curve fit.**

Assuming a $F^{14}C_{true}$ value for Fossil Ref of 0.01 and a F¹⁴C value of the modern contaminant ($F^{14}C_{add}$) of 1, the best fitting curve through the Fossil Ref samples indicates a constant level of modern contamination ($\mu gC_{add}$) of 0.5 +/- 0.1 µgC. Larger samples (>50 µgC) show an offset that can be explained with a size dependent contamination, an additional 0.1 µgC every 10 µgC collected, which we can correct for.





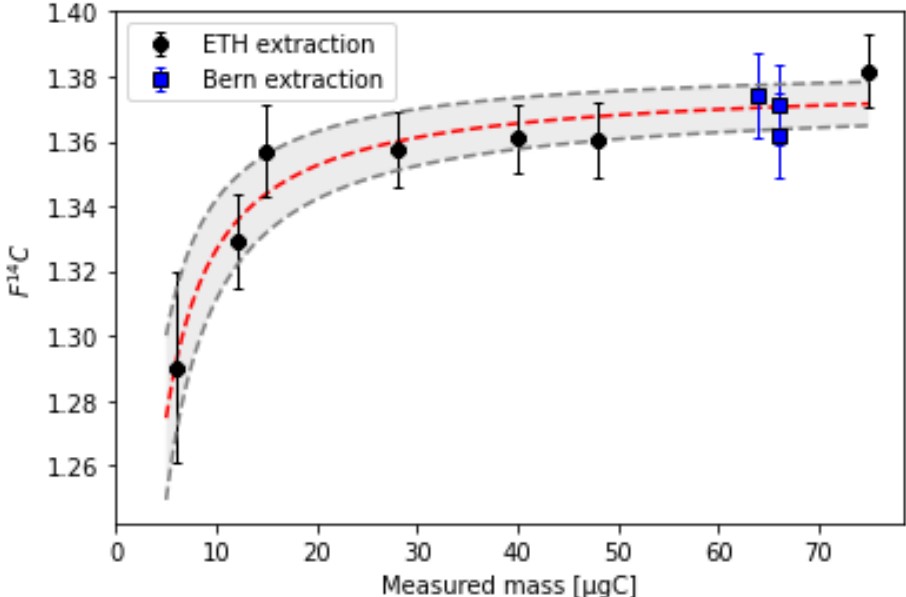

**Figure 4: Measured F$^{14}$C values against the measured mass of the samples collected from Ref 1. Symbols in blue are the Ref 1 samples extracted in Bern and are not included for the curve fitting. The grey bands represent one sigma uncertainty bar on the curve fit.**

The data collected from Ref 1 best fit onto a curve with an F$^{14}$C value of 1.38 +/- 0.01 for the reference gas. However, by considering only the quantified modern contamination of 0.5 µgC, we do not achieve the best fitting curve, and we need to

add approximately 0.2 +/- 0.1 µg of contaminant fossil carbon. Zazzeri et al. (2021) indicated that some fossil carbon might be produced within the combustion furnace, and therefore it is likely that even with our setup the combustion process led to the production of some fossil carbon.

Ref 1 samples extracted in Bern, blue markers in Fig 4 (not included for determining the constant contamination), agree well with the F$^{14}$C values for Ref 1 samples with the same mass (60 µgC) extracted at ETH, indicating that the two extraction

methods are comparable. Samples of 60 µgC are equivalent to two/three hours of sampling of ambient air with our portable system at 500 ccm or 60 liters of ambient air with the extraction line in Bern.

## 3 Discussion

In order to make the sampler portable we have reduced the size of the sampler components compared to the system in Zazzeri et al. (2021). The main changes include:

-    a smaller $CO_2$ trap placed before the combustion furnace, with 14 g against 60 g of molecular sieve. Its adsorption capacity is demonstrated by the very low modern blank, which indicates that all the ambient $CO_2$ is captured while sampling;





- a new design of the sample trap, with 0.250 g of molecular sieve against 1 g, accommodated in a 4 cm length tube and connected to a single 4 ports VICI valve. Collection of 60 µgC (two/three hours of sampling at 500 ccm) has been achieved by cooling down the sample trap using two Peltier elements;
- a smaller combustion furnace built at the Laboratory of Ion Beam Physics;
- connections and tubing of 1/8" size instead of 1/4".

All these modifications led to an important reduction of the level of constant modern contamination, from 5.5 +/- 1.1 µgC in Zazzeri et al. (2021) down to 0.5 +/- 0.1 µgC. According to the $F^{14}C$ measurements of our modern reference cylinder (Ref 1), we have an extra 0.2 +/- 0.1 µgC of fossil contamination, leading to 0.70 +/- 1.4 µgC total amount of contaminant carbon with an averaged $F^{14}C$ value of 0.71. We also found a size dependent contamination of 1%, which can be explained either with a tiny leak within the sampler or with some outgassing.

The overall uncertainty for individual samples of 60 µgC, calculated by propagating the error from counting statistics and background uncertainty, is 0.9%, comparable with other measurements techniques for $^{14}CH_4$, demonstrating that a larger sample, and therefore a longer sampling time, is not needed.

The main benefit of a portable system that needs only 60 liters of air for one sample is the important time saving both in the field and in the laboratory. The sample processing time in the laboratory has been reduced massively, and so the likelihood of contamination and mistakes by the operator. The system, given its small size, could be placed in a vehicle, enabling sampling in a source area, such as a landfill site or an urban environment, or performing a mapping of isotopic signatures in a region.

## 4    Conclusions

We have advanced the $CH_4$ sampling system from Zazzeri et al. (2021) into a portable system that can be used in field campaigns, while also reducing the contamination in the system. Further improvements could be made to automate the system, so that the valves and pumps switching, and the flow rate are computer-controlled, making the whole sampling procedure more consistent. More samples could be collected in parallel at the same time. In addition, the $CO_2$ desorbed from the sample trap is presently cryogenically trapped in glass ampules sealed for offline $^{14}C$ measurements, but a direct coupling of the zeolite trap to the gas interface (Wacker et al., 2013) connected to the MICADAS AMS system could be implemented, avoiding the additional step using glass ampules.

Full assessment of the fossil carbon contamination in $^{14}CH_4$ measurements is still challenging because there is no modern reference material available for $CH_4$. The production of a modern $CH_4$ standard for $^{14}C$ analysis, followed by an inter-laboratory comparison, should be pursued.

Overall, the combination of a selective and field deployable $CH_4$ sampler and sensitive AMS analysis provides a unique technology, that can expand the use of $^{14}CH_4$ measurements.



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



## 6 Competing interests

230 The contact author has declared that none of the authors has any competing interests.

## 7 Acknowledgement

This work has been funded by the Horizon 2020 Framework Programme (Call: H2020-MSCA-IF-2020, Project: 101026926 — FORM) and by the Laboratory of Ion Beam Physics.

235