# Peer review of "A new portable sampler of atmospheric methane for radiocarbon measurements"

_Atmospheric Measurement Techniques, 2024_

## Author Comment (AC2)

We thank the referees for their comments. Below you can find a response to each comment that has been raised.

**REF 1**

1. The introduction refers to the setup presented by Palonen et al. (2017), who built a similar portable CH4-oxidation setup to sample CH4 from respiration chambers. The authors herein conclude that the Palonen et al. setup is only usable for concentrations >100 ppm. How was this conclusion reached? While it is true that respiration chambers used in temperate climate wetlands will contain much higher CH4 concentrations, I think the setup could in theory be used for different approaches, although sampling will take a long time.

The Palonen et al. setup could be certainly used to collect samples for atmospheric $^{14}C$ measurements of atmospheric methane with some modifications, such as a longer sampling time and an increased amount of molecular sieves to scrub the atmospheric $CO_2$ from the gas stream. However, that study shows the use of the setup for high methane concentrations only. To clarify, in Line 43 we added "A portable system using a similar technique was demonstrated by Palonen et al. (2017), but in this study they used only samples with enriched methane concentrations".

2. In the method section it is stated that the setup is battery powered, which is obviously useful when working in remote areas. What type of battery was used and how long can the system run until the battery is empty?

The system has been tested only in a laboratory setting so far. However, according to our calculation, two 72V 30Ah 2160Wh Lithium batteries are sufficient to run the sampling system for 10 hours. This has been added in line 80.

3. The setup uses a Nafion membrane to remove H2O from the gas stream. Was the amount of H2O in the sample quantified? I think it would be relevant to mention because 13X zeolith, that is used to trap the CH4-derived-CO2, is also able to adsorb H2O. In addition to that, the oxidation of CH4 to CO2 will also produce H2O that will adsorb onto the sample trap.

The water level from the gas stream was not quantified, as we did not have a sensor for water measurements. However, the water adsorption has been previously tested in Zazzeri et al. 2021, demonstrating that the nafion dryer is able to reduce the water level to 0.01 %, increasing the capacity of the 13X beads in the $CO_2$ trap to adsorb atmospheric $CO_2$. A sentence about the drying capacity of the Nafion has been added in line 65.

The water derived from the $CH_4$ oxidation is adsorbed onto a magnesium perchlorate trap (water trap in the system schematic in Figure 1). This has been clarified in line 73. While extracting the sampled $CO_2$ from the sample trap, before cryogenically sealing the $CO_2$ into a glass ampule, we measured the total pressure of the gases desorbed using the calibrated volume and the pressure transducer in the TSE (Tube Sealing Equipment) system (https://www.ionplus.ch/tse). This has been clarified in line 98 of the manuscript. This would give us an approximate idea of the level of water adsorbed into the trap during sampling, and weather we needed to change the magnesium perchlorate trap placed after the furnace, due to its saturation.

4. 4. What was the reason for the catalyst that was used? Palonen et al. (2017) for example used Pd-based catalysts that yielded sufficient results at lower furnace temperatures, potentially saving more battery power.

We used the catalyst in Petrenko et al. (2008), which demonstrates that the use of such catalyst yields no detectable carbon sample memory in the line. Other catalysts could be tested for future work. We added the Petrenko et al. reference, see line 72.

5.  The assessment of contamination is valid, however, wouldn't it have been more realistic to produce a mixture of gases to test contamination as well as trapping efficiency? For example, create a gas mixture that contains fossil CH4 and Ox-II derived CO2 as a sensitive measure for sufficient separation of gases.

The use of gas mixtures would be certainly more realistic and that could be used to further test the system. Unfortunately, due to the strict schedule of the project, we decided to start first with the characterization of the modern blank using a cylinder with a 2 ppm mixture of fossil methane and synthetic air with no $CO_2$, CO or hydrocarbons (Fossil Ref), and testing the separation of gases using only a pressurized cylinder of ambient air, which was easy to retrieve. Further testing using gas mixtures could be part of future work and it has been included in the discussion section (line 184-186).

6.  6. How was the evaluation using the best-fit for contamination in Fig. 3 and 4 done? Did you use a script or automation to assess the best fit or was the fit eyeballed by manually manipulating input parameters such as mass of contamination and 14C-content?

In order to find the best fitting curve, we wrote a python script that automatically finds the best fit while reducing the chi-squared. This has been added in line 125. See the "curve_fit" function from the "scipi.optimize" package.

**REF2**

1.  Separation of atmospheric methane from CO2. In this system, the test of complete separation of methane from atmospheric methane and CO2 depends on the detection of the CO2 sensor in the system. However, the NDIR-type CO2 sensor needs a long time and strick working environment before achieving the stability and accuracy of 1 ppm, and even then, a 1 ppm error is potentially significant for an atmospheric methane concentration of 2 ppm. In addition, since the difference in F 14C values between atmospheric methane and CO2 may not be significant, the experimental results given in the article do not fully demonstrate that the system can completely separate methane and CO2 . It is recommended that the authors hold another experiment in which a 60L gas mixture is configured: 2 ppm of background CH4, 400 ppm of CO2 from combustion of OXII, and N2, or He, as the remaining gas. Perform experiments on this system using this mixture gas, and test the F14C of background methane, and the F14C of CO2.

The capability of the 13X beads to trap atmospheric $CO_2$ has been tested before implementing the trap into the system. The $CO_2$ trap was flushed with ambient air at a flow rate of 1 lpm, with the NDIR sensor measuring the $CO_2$ level after the trap. A plot of the $CO_2$ level vs time has been added in Appendix A (Figure A1). The plot shows that 14 gr of 13X beads were able to adsorb atmospheric $CO_2$ from at least 270 L of ambient air, and therefore we were quite confident that the trap was enough to separate atmospheric $CO_2$ for one sample of 60 L. To confirm the full separation, we collected 3 blanks by running the system with lab air and without combustion, each one for three hours, while also keeping track of the concentrations measured by the NDIR sensor. We did not extract any carbon from these blanks as stated in line 132. Moreover, the sensor baseline was regularly calibrated using nitrogen. Note that in the plot showing the $CO_2$ trap adsorption (Figure A1), the baseline is not 0 but negative, as the baseline had to be reset every few samplings.

The use of gas mixtures, as also suggested by the other referee, will be part of the further testing and development of the system, and we recommended that in the discussion section (see line 185).

2.  Problems with water removal in this system. Nafion dryer tubes are not very efficient at removing water, I wonder how efficient is it in this system. For 60L volume of samples with different atmospheric humidity, will the water removal efficiency of this nafion drying tube affect the absorption of CO2 by molecular sieves and the separation of CO2 from methane?

In our setup we used the same nafion dryer as Zazzeri et al. 2021 (Perma Pure gas dryer, PD-50-24), which was used to reduce the water content in ambient air to levels of 0.01 %. This has been added in line 65 of the manuscript. For this work the adsorption of atmospheric $CO_2$ by the $CO_2$ trap has been tested with the nafion drier in laboratory conditions, but the nafion drying capacity has been extensively tested at Imperial College London with different levels of air humidity both in the work of Zazzeri et al. 2021 and Saboya et al. 2022. In Zazzeri et al. 2021 the water level before the $CO_2$ trap was constantly monitored using a Picarro instrument, demonstrating that the PD model of nafion was efficiently removing the water throughout the whole sampling time. The model of the nafion dryer has been added in the manuscript in line 63.

3. Please specify in the article whether the power supply for the methane oxidize furnace at 800 degrees Celsius comes from the battery pack in the system or whether an additional power supply is required.

A sentence about the power required for whole system and the type of batteries has been included. See line 80 of the manuscript.